# Efficacy and Side Effect Profile of Intrathecal Morphine versus Distal Femoral Triangle Nerve Block for Analgesia following Total Knee Arthroplasty: A Randomized Trial

**DOI:** 10.3390/jcm11236945

**Published:** 2022-11-25

**Authors:** Marek Janiak, Rafal Kowalczyk, Grzegorz Gorniewski, Kinga Olczyk-Miiller, Marcin Kowalski, Piotr Nowakowski, Janusz Trzebicki

**Affiliations:** 11st Department of Anesthesiology and Intensive Care, Medical University of Warsaw, 02-091 Warszawa, Poland; 2Department of Anesthesiology and Intensive Care Education, Medical University of Warsaw, 02-091 Warszawa, Poland; 3Department of Orthopedics and Traumatology, Medical University of Warsaw, 02-091 Warszawa, Poland; 4Department of Anesthesiology and Intensive Care, Gruca Orthopaedic and Trauma Teaching Hospital, 05-400 Otwock, Poland

**Keywords:** knee arthroplasty, intrathecal morphine, femoral triangle nerve block

## Abstract

(1) Background: The management of postoperative pain after knee replacement is an important clinical problem. The best results in the treatment of postoperative pain are obtained using multimodal therapy principles. Intrathecal morphine (ITM) and single-shot femoral nerve block (SSFNB) are practiced in the treatment of postoperative pain after knee replacement, with the most optimal methods still under debate. The aim of this study was to compare the analgesic efficacy with special consideration of selected side effects of both methods. (2) Materials and methods: Fifty-two consecutive patients undergoing knee arthroplasty surgery at the Department of Orthopedics and Traumatology of the Medical University of Warsaw were included in the study. Patients were randomly allocated to one of two groups. In the ITM group, 100 micrograms of intrathecal morphine were used, and in the SSFNB group, a femoral nerve block in the distal femoral triangle was used as postoperative analgesia. The other elements of anesthesia and surgery did not differ between the groups. (3) Results: The total dose of morphine administered in the postoperative period and the effectiveness of pain management did not differ significantly between the groups (cumulative median morphine dose in 24 h in the ITM group 31 mg vs. SSFNB group 29 mg). The incidence of nausea and pruritus in the postoperative period differed significantly in favor of patients treated with a femoral nerve block. (4) Conclusions: Although intrathecal administration of morphine is similarly effective in the treatment of pain after knee replacement surgery as a single femoral triangle nerve block, it is associated with a higher incidence of cumbersome side effects, primarily nausea and pruritus.

## 1. Introduction

Total knee arthroplasty is a common surgical procedure with an expected increase in the number of cases due to the aging of populations and the increase in obesity [1]. Over 33,000 procedures were performed in Poland in 2019, and the number has been rising since the year 2005, as seen in data from the Polish Central Endoprostheses Database of the National Health Fund [2].

Postoperative pain following total knee arthroplasty is described as moderate to very severe by most patients [3]. It is well documented that postoperative multimodal analgesic therapy using combinations of regional analgesic techniques provides the most optimal postoperative pain control [4], but direct comparisons of different regional blocks do not provide a definitive answer to which technique is most preferred. To date, no one regional anesthetic technique is recommended above all others for postoperative pain control following total knee arthroplasty [5,6]. Among methods used to combat severe postoperative pain, both intrathecal morphine and femoral nerve blocks have been used. Both methods are simple to perform and have been shown to be effective in pain control but have a differing profile of side effects and reported patient satisfaction [4]. The aim of this study was to compare the analgesic efficacy of intrathecal morphine versus a single-shot distal femoral triangle nerve block with a special focus on the incidence of side effects related to morphine use.

## 2. Materials and Methods

In line with the Helsinki Declaration, the Bioethical Committee of the Medical University of Warsaw approved the study (number KB/107/2016, Chairperson Prof. Zbigniew Wierzbicki). Consecutive patients scheduled for primary total knee arthroplasty in the Orthopedic Department of the Medical University of Warsaw were included in the trial. All patients meeting the inclusion criteria received information materials prior and were asked for formal consent to participate in the trial. The exclusion criteria were: lack of consent for inclusion in the trial, lack of consent or a contraindication to performing the regional block techniques employed in the trial, American Association of Anesthesiology score (ASA) of IV or V, chronic opioid use, allergy or contraindications to the drugs used in the trial such as paracetamol, metamizole or ketoprofen.

The trial participants were anesthetized in a block room using a standardized procedure, as per the routine used for knee arthroplasty in our clinical center. Mandatory basic parameters, including an electrocardiogram trace, pulse-oximetry and noninvasive blood pressure, were monitored throughout from the time of admission to the surgical theater area to discharge from the postoperative recovery unit. Allocation to either the intrathecal morphine or distal femoral triangle nerve block was performed using a randomized list known to only one trial coordinator that was not performing the blocks. Due to obvious reasons, double-blinding of the procedure was not fully possible. All the procedures were performed under direct supervision of experienced anesthesiologists. Prior to the anesthetic block, all patients received an intravenous premedication with 0.1 mg of fentanyl (Fentanyl WZF, Polpha, Warszawa, Poland) and 2 mg of midazolam (Midanium WZF, Polpha, Warszawa, Poland). Participants randomized to the intrathecal morphine group (ITM) had a spinal block in the sitting position with 15 mg of 0.5% hyperbaric bupivacaine (Marcaine 0.5% Heavy Spinal, Aspen Pharma Trading Ltd., Dublin, Ireland) and 100 micrograms of intrathecal morphine (Morphini Sulfas WZF 0.1% Spinal, Polfa, Warszawa, Poland). An aseptic technique was used for the intrathecal block in the L3/L4 vertebral interspace using a 26 G atraumatic spinal needle (Atraucan, B. Braun Melsungen AG, Melsungen, Germany). Participants in the single-shot femoral nerve block group (SSFNB) had a distal femoral nerve block within the femoral triangle performed using an aseptic technique under ultrasound guidance with a high linear frequency (12–15 MHz) probe and an 80 mm echogenic block needle (Stimuplex Ultra 360, B. Braun Melsungen AG, Melsungen, Germany). A dose of 20 mL of 0.25% bupivacaine with adrenaline (original solution Marcaine Adrenaline 0.5%, Aspen Pharma Trading Ltd., Dublin, Ireland) was administered on confirmation of sub-sartorial spread lateral to the femoral artery, just as it dives under the sartorius muscle. On confirmation of sensory block with a decrease in sensation to cold in the front peripatellar thigh region, an intrathecal block was performed in the same way as the ITM group, but no intrathecal morphine was administered. Participants in both groups had their spinal block assessed using the Bromage scale and, on confirmation of spinal block effectiveness, were transferred to the operating theater. 

The surgical procedure was performed by the same surgical team comprising two orthopedic specialists, with a standardized surgical procedure using a medial peripatellar approach, sacrificing the cruciate ligaments and using bone cement for prosthesis fixation. In all cases, a tourniquet was used to optimize surgical conditions and reduce intraoperative blood loss which was deflated prior to wound closure. Antimicrobial perioperative prophylaxis and thromboprophylaxis with low molecular weight heparins were implemented in all participants of the study as per hospital protocol.

Monitoring of vital parameters was continued throughout the surgical procedure with intravenous fluid therapy given at the discretion of the anesthesiologist, and in individual cases of patient discomfort, moderate sedation with propofol was used. Following the surgical procedure, all patients were transferred to the postoperative care unit (PACU), where they were monitored for a 24 h period, after which they were discharged to the orthopedic ward. Postoperative analgesia was standardized, with regular intravenous paracetamol 1 g every 6 h and ketoprofen 100 mg every 12 h. All patients had rescue morphine at 0.1 mg/kg administered intravenously on demand under nurse-controlled analgesia every 6 h whenever the Numerical Rating Score (NRS) was more than 4. Vital parameters were recorded every hour. Additionally, trial participants were asked to assess their pain and side effects, such as nausea, vomiting and pruritus, at 1, 6, 2, 24, 48 and 72 h following surgery. At these time points, the nurse also recorded vital parameters and sedation levels. If required, additional doses of morphine and intravenous ondansetron 4 mg were administered. Pain was assessed at all time points using the Numerical Rating Score (NRS) from 0 (no pain) to 10 (worst possible pain) both at rest and with active knee flexion of the operated side. Urinary retention was not assessed as study participants had urinary catheterization. 

Statistical analysis of the obtained data was performed using Statistica 13.1 (StatSoft Inc., Tulsa, OK, USA). The data is described using mean values and standard deviations as a measure of dispersion in the case of continuous values or cumulative values for non-continuous data. Comparison of measured variables between groups was performed using the student *t*-test for the parametrical data and the U Mann–Whitney test for the non-parametric data. A normality test by Kolmogorov–Smirnov was performed. For non-continuous data sets, the Chi2 test was used to compare variables. A statistically significant value of *p* < 0.05 was used. A post-hoc power analysis showed the power to be >90% for most variables, such as nausea or pruritus, when considering the sample size.

## 3. Results

A total of 52 participants were enrolled in the study, with 26 per group. The two groups did not differ in their basic characteristics such as sex, age, anthropometry, ASA classification or duration of surgery, as can be seen in Table 1.

Table 2 shows the results of the assessed variables between the ITM and SSFNB groups. The cumulative morphine dose did not differ between the two groups in the 72 h observation period. A statistically and clinically relevant reduction in nausea and pruritus could be seen in the postoperative period in the femoral triangle nerve group. Both pruritus and nausea occurred in at least half of the group that received the intrathecal morphine but were a rare occurrence in the femoral nerve block group. More patients required the administration of ondansetron in the postoperative period in the ITM group, and this was statistically significant. The effectiveness of the analgesic regimen did not differ between the two groups in terms of the NRS results (Table 3 and Table 4). A benefit of the femoral nerve block was noted at some time points, such as at six postoperative hours at rest (*p* = 0.0361) and on discharge from the PACU when NRS was assessed on knee flexion (*p* = 0.0138).

No relevant complications of the spinal block or the femoral nerve block were noted among the study participants, but two patients required the administration of a small dose of naloxone due to bradypnea with overt sedation in the ITM group.

Intra- and postoperative hemodynamic values of blood pressure, heart rate or oxygen saturation did not differ between the two groups. With the exception of two cases requiring naloxone administration, no desaturations relevant to the study were noted, but it must be stated that the measurements were recorded at specific time points, and any reduced value of oxygen saturation was treated with oxygen supplementation, which was not recorded.

## 4. Discussion

We assessed the efficacy of a 100 mcg intrathecal morphine in comparison to a single-shot femoral nerve block performed in the femoral triangle with a sub-sartorial spread just lateral to the femoral artery for postoperative analgesia following total knee arthroplasty. Our study shows that both these methods are equianalgesic and can be used alternatively. However, the undesirable side effect profile of intrathecal morphine must be taken into account, with a potential for a rare but dangerous respiratory depression. 

Total knee arthroplasty is an orthopedic procedure commonly performed for gonarthrosis. Although the aim of the surgery is to reduce chronic pain related to knee degeneration [7], pain intensity can be very severe directly after the surgery. Effective analgesia with the use of regional blocks is optimal for fast-track patient mobilization and achievement of good functional recovery of the knee joint [8,9]. It is possible that the use of regional anesthetic techniques could help reduce hospital stay time and the incidence of side effects related to long-term opioid use [10].

A single-shot femoral nerve block is one of the accessible methods of pain management following knee arthroplasty [11]. It does not provide prolonged analgesia with the added flexibility that a continuous femoral triangle nerve block can [12,13], especially when combined with a sciatic nerve block [14], but in comparison to the continuous nerve block, it carries a reduced risk of falls in the postoperative recovery time [15,16]. A study by Wyatt et al. [17] showed no major advantage of a continuous femoral nerve block over a single-shot technique when used in combination with ITM. For these reasons, a single-shot technique with a more distant block area and no sciatic nerve blocking was chosen in our trial to reduce the risk of a fall due to quadriceps muscle weakness in early mobilization. However, we do note that the most recent PROSPECT guidelines on total knee arthroplasty do not recommend any femoral nerve block due to the potential negative impact on functional, fast-track recovery [18], but the study was designed in the time before the focus was placed on the higher risk of falls following total knee arthroplasty and this may still be debatable. No study participant experienced a fall, and femoral nerve block is still used as an analgesic option as local infiltration analgesia (LIA) is not practiced by our orthopedic surgeons. 

Single-shot femoral nerve block following total knee arthroplasty is more effective compared to simple local wound infiltration [19]. It has a similar or less effective profile when compared to an adductor canal block which, on the other hand, helps preserve more motor function of the quadriceps muscle of the thigh [20,21]. A shift toward finding more optimal distal motor-sparing blocks is observed in the literature. A novel parasartorial compartment (PASC) block is one such promising alternative [22]. The study by Lee et al. [12] found a comparable analgesic result when a continuous catheter was placed in the femoral triangle in comparison to a proximal and distal adductor canal catheter position. 

Morphine administered intrathecally (ITM) has a proven efficacy in the treatment of postoperative pain following large joint arthroplasties [23]. The benefit of ITM is the ease of administration and no additional complications related to the injection itself as compared to the more invasive nerve block procedure. The major drawback of ITM is the high incidence of side effects that are not well tolerated by patients, such as nausea, pruritis and sedation [24]. A potential complication remains late apnea related to post-opioid respiratory center depression [25]. In our study, 2 participants in the ITM group experienced respiratory depression with bradypnea and overt sedation after 15 h and required administration of naloxone with full recovery. No evident drop in saturation was noted as oxygen was being administered, but the apnea triggered a monitor alarm. This is not statistically significant as the study groups are small, but it remains clinically relevant. As the cumulative systemic morphine dose did not differ between the groups, it should be noted that ITM may have caused respiratory depression. No additional factors, such as obstructive sleep apnea, were found to contribute to these cases. The patients remained in the PACU for the 24 h postoperative period and were discharged to the ward with no further action required. 

In both our study groups, undesirable side effects were observed secondary to the implemented analgesia. The incidence of nausea and pruritis was significantly higher in the ITM group, even though the overall opioid consumption was similar in both groups. Our trial results remain in line with a metanalysis which showed an associated increased pruritis and a decreased patient satisfaction [26,27], although no difference in side effect profile was observed in one study [28]. 

Recommendations pertaining to analgesia for total knee arthroplasty combine paracetamol, non-steroidal anti-inflammatory drugs and opioid therapy [6,9,18]. The used multimodal analgesic regimen in the trial provided good pain management in both study groups.

Our study has several limitations. Firstly, the study was not blinded—both the anesthesiologist performing the block and the patient were aware of the group allocation. However, the personnel assessing the outcomes, including pain scores and side effects, were unaware of the group allocation of the participants. Secondly, our study investigates a femoral triangle nerve block which may be related to quadriceps muscle weakness affecting early mobilization. We did not assess patient satisfaction in the perioperative period.

## 5. Conclusions

In our randomized trial, the results show a similar overall efficacy of intrathecal morphine at a dose of 100 micrograms to a single-shot femoral triangle nerve block but with a higher incidence of undesirable side effects among patients receiving intrathecal morphine, especially nausea and pruritus. The risk of respiratory depression, which occurred in our study in the intrathecal morphine group, confirms the need for respiratory monitoring and limits its possible safe use in day-case surgery.

## Figures and Tables

**Table 1 jcm-11-06945-t001:** Patient characteristics.

	ITM Group	SSFNB Group	*p* Value
Sex Female/Male (%)	23/3 (88.5%/11.5%)	23/3 (88.5%/11.5%)	*p* = 1 (Chi2)
Age (years)	68 +/− 11.9	67.5 +/− 9.7	*p* = 0.86 (*t* test)
Height (cm)	161.2 +/− 6.4	161.5 +/− 6.4	*p* = 0.85 (*t* test)
Weight (kg)	82.2 +/− 15.1	81 +/− 16.6	*p* = 0.78 (*t* test)
Surgical procedure time (min)	87.6 +/− 17.7	92.9 +/− 29.4	*p* = 0.43 (*t* test)
Tourniquet time (min)	72.5 +/− 11.9	73.1 +/− 21.9	*p* = 0.9 (*t* test)
ASA 1/2/3 (%)	1/25/0 (3.8%/96.2%/0)	0/24/2 (0/92.3%/7.7%)	*p* = 0.36 (Chi2)

Values are presented as mean +/− SD or as number/percentage. ASA—American Society of Anesthesiology physical status scale. ITM = intrathecal morphine, SSFNB = single-shot femoral triangle nerve block.

**Table 2 jcm-11-06945-t002:** Treatment results.

	ITM Group	SSFNB Group	*p* Value (Statistical Test)
Cumulative morphine dose (mg)	31 [23–37]	29 [23–31]	*p* = 0.26 (U Mann–Whitney)
Nausea N (%)	13 (50%)	2 (7.7%)	*p* = 0.0008 (Chi2)
Vomiting N (%)	7 (26.9%)	1 (3.8%)	*p* = 0.211 (Chi2)
Pruritus N (%)	14 (53.8%)	1 (3.8%)	*p* = 0.0001 (Chi2)
Somnolence N (%)	15 (57.7%)	9 (34.6%)	*p* = 0.09 (Chi2)
Maximum NRS at rest	4 [2–5]	2 [0–6]	*p* = 0.18 (U Mann–Whitney)
Maximum NRS on motion	4 [3–7]	3.5 [2–7]	*p* = 0.22 (U Mann–Whitney)
Number of patients requiring ondansetron N (%)	13 (50%)	3 (11.6%)	*p* = 0.0271 (Chi2)
Number of patients requiring naloxone N (%)	2 (7.7%)	0 (0%)	*p* = 0.1649 (Chi2)

ITM = intrathecal morphine. SSFNB = single-shot femoral triangle nerve block. Values x[y-z] signify median[interquartile range].

**Table 3 jcm-11-06945-t003:** Postoperative pain levels at rest.

NRS at Rest	ITM Group	SSFNB Group	*p* Value (U Mann–Whitney)
On admission to PACU	0 [0–0]	0 [0–0]	*p* = 1
At 3 h	0 [0–0]	0 [0–0]	*p* = 0.7418
At 6 h	3 [0–4]	0 [0–2]	*p* = 0.0361
At 9 h	0 [0–2]	1 [0–3]	*p* = 0.602
At 12 h	1 [0–2]	0 [0–2]	*p* = 0.3554
At 18 h	0.5 [0–2]	0 [0–0]	*p* = 0.07
On discharge from PACU	0 [0–1]	0 [0–0]	*p* = 0.1938

NRS = numerical Rating Score, ITM = intrathecal morphine, SSFNB = single-shot femoral triangle nerve block.

**Table 4 jcm-11-06945-t004:** Postoperative pain levels on motion.

NRS on Motion	ITM Group	SSFNB Group	*p* Value (U Mann–Whitney)
On admission to PACU	0 [0–0]	0 [0–0]	*p* = 0.819
At 3 h	0 [0–0]	0 [0–0]	*p* = 0.7007
At 6 h	3.5 [0–5]	0 [0–3]	*p* = 0.0582
At 9 h	3 [0–4]	2 [0–4]	*p* = 0.7143
At 12 h	2.5 [1–3]	1 [0–4]	*p* = 0.1176
At 18 h	1.5 [1–4]	1 [0–2]	*p* = 0.1242
On discharge from PACU	1 [1–4]	0 [0–2]	*p* = 0.0138

NRS = numerical Rating Score, ITM = intrathecal morphine, SSFNB = single-shot femoral triangle nerve block.

## Data Availability

The data presented in this study may be available on request made to the corresponding author.

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
