# Peer review of "Efficacy and Side Effect Profile of Intrathecal Morphine versus Distal Femoral Triangle Nerve Block for Analgesia following Total Knee Arthroplasty: A Randomized Trial"

_jcm, 2022, doi:10.3390/jcm11236945_

Round 1
Reviewer 1 Report
I would like to congratulate the authors on this well written paper.
The only comment is on the importance of the duration of analgesia.
The duration of analgesia is best achieved by a continuous nerve block, albeit to a certain degree by intrathecal morphine (ITM). Although the single shot block may offer similar analgesia compared to ITM, the duration of analgesia may be inferior to ITM and continuous nerve block. This may be worth mentioning.
Author Response
We would like to thank the reviewer for the support of this manuscript.
We do agree that continuous nerve block techniques prolong the duration of analgesia in an effective manner and can provide additional flexibility in postoperative pain therapy. However, the increased risk of cathether dislodgement, infection and additional costs with higher personnel demands may make it suboptimal in our case study. Also, a continuous block may interfere with goals set by early mobilization.
This is now covered in the third paragraph of our discussion. Point [12,13] of reference.
Reviewer 2 Report
This was a very well-written manuscript. More studies that compare the efficacy of various techniques are needed, so this is a welcomed attempt at doing so. Although the findings aren't too surprising, it is important to have this type of literature out there to support or refute the various practices.
That being said, there are a few suggestions. It is important to reference and discuss other similar work that has been done in this space. For example, the paper by Olive et al 2015 (PMID 26099756) or Wyatt et al 2015 (PMID 25653286) or Toolyodpun (PMID 36260155). I would also focus on how this particular study differentiates itself from those that have already been performed.
The third paragraph is a lengthy discussion on falls, which isn't the primary aim of interest, but rather a side effect that has arguable been reviewed in numerous papers. I would probably use this text space to focus on the analgesic efficacy of the approaches.
Lastly, the authors mentioned respiratory depression, but did not quantify what that entailed. Was it desaturation? If so, by how much? Were they bumped to a higher level of care? Any added treatments? Did the patients have elevated risks of respiratory depression with specific comorbidities (e.g. OSA).
Author Response
This was a very well-written manuscript. More studies that compare the efficacy of various techniques are needed, so this is a welcomed attempt at doing so. Although the findings aren't too surprising, it is important to have this type of literature out there to support or refute the various practices.
Our response: We would like to thank the reviewer for his kind support.
That being said, there are a few suggestions. It is important to reference and discuss other similar work that has been done in this space. For example, the paper by Olive et al 2015 (PMID 26099756) or Wyatt et al 2015 (PMID 25653286) or Toolyodpun (PMID 36260155). I would also focus on how this particular study differentiates itself from those that have already been performed.
Our respone: We have considered adding into the discussion similar trials. However, the Toolyodpun (PMID 36260155) trial had all patients receiving a local infiltration analgesia (LIA) which is not performed in our centre and may cause a differing analgesic effect - this is mentioned in the discussion. Olive et al (PMID 26099756) includes patients on a continuous femoral nerve block vs ITM vs a combination. A continuous cathether technique may interfere with early mobilization goals - this is mentioned in the third paragraph of our discussion. Wyatt et al. (PMID 25653286) actually concludes that a continuous technique has no significant advantage over a single shot when used in combination with ITM - we have added this into our discussion.
The third paragraph is a lengthy discussion on falls, which isn't the primary aim of interest, but rather a side effect that has arguable been reviewed in numerous papers. I would probably use this text space to focus on the analgesic efficacy of the approaches.
Our response: We fully agree with the reviewer that the incidence of falls is not related to our aims and have reduced the word count of the discussion. We have not removed the paragraph entirely, as it also discusses continuous nerve techniques and readers of this article should keep in mind the risk of falls. We hope this will be satisfactory to the reviewer.
Lastly, the authors mentioned respiratory depression, but did not quantify what that entailed. Was it desaturation? If so, by how much? Were they bumped to a higher level of care? Any added treatments? Did the patients have elevated risks of respiratory depression with specific comorbidities (e.g. OSA).
Our response: We agree with providing more detail on these incidents and have added the information in the discussion. The patients did not present relevant desaturation, but an apnea alarm in the monitor was trigerred with overt sedation. The patients remained in the PACU for the rest of the 24 hour period and were safely discharged to the ward. No additional contributing factors for the event were identified.
Reviewer 3 Report
Dear Authors,
i congratulate with you for your interesting work.
In my opinion, you should mention in the discussion a new interesting modified femoral-triangle block, the "para-sartorial compartments block", which is a new promising technique aimed to enhance the efficacy of femoral triangle block. I suggest you to cite in in your discussion : Pascarella G, Costa F, Del Buono R, Strumia A, Cataldo R, Agrò F, Carassiti M. The para-sartorial compartments (PASC) block: a new approach to the femoral triangle block for complete analgesia of the anterior knee. Anaesth Rep. 2022
Author Response
Dear Reviewer,
We would like to Thank You for the review of our manuscript.
We fully agree with parasartorial blocks being the new kid on the block in multimodal analgesia for total knee arthroplasties and this was not yet the situation at the start of our trial. We have incorporated the cited article into the discussion due to its high relevance.